# Non-O1/O139 environmental *Vibrio cholerae* from Northern Cameroon reveals potential intra-/inter-continental transmissions

**Deborah Yebon Kang**[1], **Mohammad Tarequl Islam**[2], **Roméo Wakayansam Bouba**[3], **Zoua Wadoubé**[3], **Moussa Djaouda**[4,5], **Yann Felix Boucher**[1,6,7] *

**1** Singapore Centre for Environmental Life Sciences Engineering (SCELSE), National University of Singapore, Singapore, **2** Infectious Diseases Division, International Centre for Diarrheal Disease Research,Dhaka, Bangladesh, **3** Department of Biological Sciences, Faculty of Science, University of Maroua, Maroua, Cameroon, **4** Department of Life and Earth Sciences, University of Maroua, Maroua, Cameroon, **5** Department of Technical and Fundamental Studies, University of Garoua, Garoua, Cameroon, **6** Saw Swee Hock School of Public Health and National University Hospital System, National University of Singapore, Singapore, **7** Infectious Diseases Translational Research Program, Department of Microbiology and Immunology, Yong Loo Lin School of Medicine, National University of Singapore and National University Hospital System, Singapore

* ephyb@nus.edu.sg

## Abstract

Northern Cameroon in Central Africa has experienced recurring cholera outbreaks despite ongoing efforts to control the disease. While most cholera studies focus on O1 pandemic *Vibrio cholerae* strains, non-O1/O139 strains are increasingly recognized for their infection potential and dynamic relationships with O1 strains during outbreaks. Here we explore the genetic diversity and phylogenetic relationships of non-O1/O139 *V. cholerae* (NOVC) isolated from environmental water sources in Northern Cameroon. These NOVC strains show significant genetic diversity and virulence potential. They are closely related to environmental strains from Kenya and clinical strains from Argentina and Haiti, suggesting transmissions across countries and continents, likely facilitated by human carriers. The highly conserved *tcpA* gene found in some strains from Cameroon is closely related to the *tcpA* O1 Classical type, suggesting direct or indirect genetic interactions between these environmental NOVC strains and pandemic strains. Our findings underscore the importance of environmental surveillance and further studies of NOVC strains to better understand cholera outbreaks.

## Author summary

Cholera is an acute diarrheal disease caused by *Vibrio cholerae*, which primarily spreads through contaminated water. It remains a global health issue, especially in regions with poor sanitation and limited access to clean drinking water, such as Sub-Saharan Africa. Although a lot of research has been done, it is still unclear how new pathogenic strains emerge from the environment and persist in endemic areas. Most studies focused on O1 and O139 serogroups associated with pandemic outbreaks, leaving a knowledge gap

**Data availability statement:** This Whole Genome Shotgun project has been deposited at DDBJ/ENA/GenBank under the BioProject accession PRJNA1108781.

**Funding:** MD's sampling effort was supported by the International Foundation for Science (IFS) through a grant (I2-W5581-2). DYK's work is supported by National University Singapore Integrative Sciences and Engineering Programme (ISEP). The funders had no role in study design, data analysis, decision to publish, or preparation of the manuscript.

**Competing interests:** The authors have declared that no competing interests exist.

about non-O1/O139 strains and their role in cholera transmission and genetic evolution. In this study, we conducted genomic analysis of non-O1/O139 *V. cholerae* isolated from environmental water sources in Northern Cameroon. Our findings reveal that these strains are genetically diverse and closely related to environmental and clinical strains from Kenya and Argentina, suggesting transmissions across different geographical regions, possibly through human carriers. The presence of conserved virulence genes, including the *tcpA* gene closely related to the O1 Classical type, indicates their potential to cause infections. This study highlights the importance of expanding cholera surveillance to include non-O1/O139 strains, as they may play a bigger role in cholera transmission than previously thought.

## Introduction

Cholera, a diarrheal disease caused by *Vibrio cholerae*, remains a global health concern, affecting nearly three million individuals annually and resulting in more than 95,000 deaths worldwide [1]. Among 69 endemic countries, Sub-Saharan Africa bears the highest burden, with nearly half of the global cases reported each year [2,3]. The Lake Chad Basin, including areas of North Cameroon, Northeast Nigeria, South Chad, and Southeast Niger, is a major endemic zone for cholera on the continent. Since the emergence of cholera in this region in 1971, there has been a net increase in epidemic events, notable from 1991 and particularly during the period from 2003 to 2010 [3–5].

Of main concern are the districts in the northern region of Cameroon, including Maroua and the surrounding areas along Lake Chad (Mora, Mada, Kousseri), the Logone River, the borders with Chad (Maga, Guéré, Velé, Yagoua, Moutourwa, Figuil, Bibémi, Pitoa), and the border with Nigeria (Kolofata, Mogodé, Bourha, Hina, Tokombéré, Guider, Mayo-oulo) [6]. This area is regularly affected by recurring cholera epidemics with prolonged duration, with nine reported epidemics between 1971 and 2015 in Cameroon. Historically, *V. cholerae* O1 El Tor biotype has been responsible for cholera epidemics in the region [7,8]. Kaas et al proposed that the pathogenic *V. cholerae* O1 strains in the region are endemic to the Lake Chad basin and distinct from other African strains [7].

As Northern Cameroon is a recognized cholera epicenter, epidemiological studies have explained several factors that facilitate cholera transmission in this region (9). These include poor access to drinking water and sanitation facilities [9], forcing the population to use water of uncontrolled quality from wells and streams. Traditional latrines and open defecation practices, coupled with periodic floodings that pollute these water sources, are regularly observed in the region [10]. The disease transmission is further exacerbated by weekly markets, ritual celebrations, and conflicts accompanied by mass movements of refugees. Although vaccination efforts have notably reduced the spread of outbreaks, several challenges persist, such as delays in accessing care in remote areas due to poor infrastructure and insufficient resources for case management [6,11].

While most studies on cholera focus on clinical strains during pandemics and epidemics, some have sought to understand the diversity of environmental *V. cholerae* and their contribution to cholera outbreaks. Environmental strains show significant heterogeneity, with the majority belonging to the non-O1/O139 *V. cholerae* (NOVC) group, often lacking a link to cholera transmission chains [12,13]. While some O1 strains in aquatic environments during cholera epidemics show clear connections to clinical cases [14–17], recent research has revealed intriguing findings about NOVC strains circulating in the environment, which are often neglected due to the predominant attention on toxigenic O1 and O139 strains.

Notably, some non-toxigenic environmental NOVC strains carry the toxin co-regulated pilus (*tcpA*) gene, a key virulence factor for colonizing the human intestinal tract and have been readily detected in cholera endemic areas [17–19]. Another interesting observation was the co-circulation of non-pandemic NOVC lineages with the pandemic lineage during large cholera outbreaks in Argentina and in Haiti [20,21]. Furthermore, some studies have shown the virulence potential of NOVC strains, indicating increased infection risk with the effect of climate change [22–24]. These findings suggest that NOVC strains may play a bigger role in infections and cholera outbreaks than previously thought and might become more prevalent on a warmer planet.

Also of concern are environmental strains of *V. paracholerae*, the closest sister species of *V. cholerae*, often misclassified as its better known relative due to their high genetic and phenotypic similarity. *V. paracholerae* has unique metabolic capabilities, such as utilizing α-cyclodextrin and pectin as sole carbon sources [25]. It harbors virulence factors like *hlyA*, *toxR*, or *ompU*, as well as mobile integrative conjugative element (ICE) commonly found in *V. cholerae*, highlighting its potential pathogenicity [25,26]. Although only recently described, it has been associated with human infections as early as the World War One [27].

In this study, we isolated 19 *V. cholerae* and one *V. paracholerae* from various drinking and bathing water sources in Northern Cameroon. The primary objectives were to determine whether there are distinct connections among environmental NOVC strains from different geographical locations and whether these strains are potentially linked to cholera transmission chains. To address these questions, we conducted a phylogenetic analysis of environmental strains from Cameroon with known reference strains from clinical and environmental sources. For closely related strains, we compared their core genome multilocus sequence type (cgMLST) and virulence factors. Our genetic virulence profile and phylogenetic analysis showed a close relationship between these Cameroon strains and environmental strains from Kenya, as well as clinical strains from Argentina, suggesting both intra-African and intercontinental spread of NOVC strains. The presence of highly conserved *tcpA* genes in NOVC strains similar to the classical type *tcpA* suggests their infection potential and ability to spread regionally and globally. These findings highlight the need of environmental surveillance in the Northern Cameroon region to monitor *tcpA*+ NOVC strains and to better understand their importance as a causative agent of diarrheal disease.

## Methods

### Ethics statement

This study did not involve human or animal subjects that require ethical approvals. The water samples were collected from natural water sources with permission from local authorities, following local regulations.

### Study site

This study was conducted in North Cameroon, a region located between latitude 9°-13° north and longitude 13°-15° east. The region is made up of heterogenic soils and has a tropical climate with two seasons: a dry season from October to April and a rainy season from May to September, with the highest rainfall in July and August. Annually, rainfall ranges from 800 to 900 mm. Temperature shows a wide fluctuation, with minimum around 18°C in January and maximum up to 42°C in March. The region is characterized by a dense hydropgraphical network made up of 'mayos'—seasonal streams that dry out during dry season—and permenant rivers traversing the area. Potable water scarcity is a significant problem, as most inhabitants rely on wells and streams for drinking and other domestic purposes.

## Water sample collection and isolation of *V. cholerae*

Water samples were collected in 2018 and 2021 from selected areas heavily affected by the major cholera outbreaks of 2010-2011 (S1 Fig). Sampling criteria included the geographical location and the availability of water sources such as streams, ponds, wells, and boreholes. Water sources coordinates were recorded using a GPS receiver. At each sampling point, water was collected in a sterile glass bottle and 300mL of water was filtered through a sterile 47 mm, 0.22 μm-pore-diameter gridded membrane filter under partial vacuum. The filter was then transferred using sterile forceps to alkaline peptone water (APW) for enrichment at 37°C for 8 hours. After incubation, a loop of the enrichment culture was streaked onto thiosulfate citrate bile salt sucrose (TCBS) agar and incubated at 37°C for 18-24 hours. Presumptive colonies (yellow, measuring 2-4 mm) were subcultured on brain heart infusion (BHI) agar to obtain pure cultures. Gram-negative, curved and motile rods that were oxidase positive were further analyzed by API 20E kit (Bio Merieux SA, France). The isolated colonies were then made into stab cultures and sent in two separate batches to Canada and Singapore for sequencing.

## DNA extraction & sequencing

Upon arrival, each strain was isolated from the stab culture by streaking onto TCBS agar. A single colony was then inoculated into 5mL of tryptic soy broth (TSB), which was incubated at 37°C at 220rpm overnight. A 0.5mL aliquot was preserved as a 20% glycerol stock, while another 0.5mL was used for DNA extraction using Qiagen DNeasy Blood & Tissue DNA extraction kit (Qiagen, Venlo, The Netherlands), following the manufacturer's protocol. To confirm the presence of *V. cholerae* and *V. paracholerae*, *viuB* PCR was performed using primers described in a prior study (28). 16S rRNA was also amplified using universal primers 27F (5'-AGAGTTTGATCMTGGCTCAG-3') and 1492R (5'-GGYTACCTTGTTACGACTT-3'), followed by Sanger sequencing and analysis using NCBI BLAST web-based tool [29,30]. Strains identified as *viuB*-positive were selected for whole genome sequencing.

Samples were sequenced at different laboratories using different NGS library preparation kits due to different timing of sample collection and the involvement of new researchers with expertise in sample analysis who joined the project during its second half. The first batch, collected earlier, was sequenced in Canada, with libraries prepared using Nextera XT DNA (Illumina, San Diego, CA, USA) and then sequenced using an Illumina MiSeq sequencing platform, with 250 bp paired-end reads. The second batch, collected later, was sequenced at the Singapore Centre for Environmental Life Sciences Engineering (SCELSE), Singapore. For this batch, library preparation was done using TruSeq Nano DNA (Illumina, San Diego, USA) with dual barcoded TruSeq DNA UD Indexes (Illumina, San Diego, USA). Sequencing was performed on an Illumina HiSeqX platform, generating 150 bp paired-end reads.

## Genome assembly and species identification

From raw reads, low-quality reads were trimmed using Trimmomatic v0.39 [31]. Genomes were assembled using Shovil pipeline (https://github.com/tseemann/shovill), with contigs equal or less than 200 bp filtered out. Gene annotation was done via Prokka v1.14.5 [32]. Species classification was first done by average nucleotide identity (ANI) calculation using fastANI v1.33 [33]. Further differentiation between *V. cholerae* and *V. paracholerae* was done through digital DNA-DNA hybridization (dDDH) (https://ggdc.dsmz.de/ggdc.php), with 70% as species threshold [34]. Antibiotic resistance genes and sequence type (ST) were identified using STARAMR v0.5.1 [35]. For *viuB* typing, a 272 bp variable region within the 816 bp gene was used for genotyping, as described in a previous study [28]. The *viuB* gene was first extracted using gene annotation information and then compared against a customized *viuB*

database (available upon request) using Geneious software. The gene was considered as the same genotype if all nucleotides were identical (perfect match) or if there was only a single nucleotide mismatch within the 272 bp region compared to the reference database.

Assembly quality and STARAMR results are available in S1 **and** S2 Tables. This Whole Genome Shotgun project has been deposited at DDBJ/ENA/GenBank under the BioProject accession PRJNA1108781.

## Comparative genome analysis

Genomes were visualized using CGView Comparison Tool v.2.0.3 [36], with *V. cholerae* El Tor N16961 as reference. To identify virulence factors, blast score ratio (BSR) was employed, as described in Rasko et al [37]. Using N16961 as the reference, we extracted virulence genes from five pathogenicity islands: CTX-Phi cluster, Vibrio Pathogenicity Island-I (VPI-I), Vibrio Pathogenicity Island-II (VPI-II), Vibrio seventh pandemic island-I (VSP-I), and Vibrio seventh pandemic island-II (VSP-II). The genes were extracted using the locus tags and subsequently blasted against the genomes of interest, using BLAST+ v2.15.0 tblastn [38]. A BSR score was calculated using the following equation:

$$BSR\ score\ = \frac{BLAST\ raw\ score\ of\ query\ vs.\ reference}{BLAST\ raw\ score\ of\ reference\ vs.\ reference}$$

Based on the BSR score, a gene was considered highly identical if the BSR was 0.8 or higher, present with sequence divergence if the BSR ranged from 0.4 to 0.8, or absent if the BSR was less than 0.4 (S3 Table) [39]. A heatmap was drawn to visualize these results using Complex-Heatmap v 2.15.4 package in R 4.3.1 [40].

## Phylogenetic analysis

For core genome analysis of *V. cholerae* and *V. paracholerae*, the core genome was extracted and aligned using Roary v3.11.2 [41] with minimum percentage identity for blastp (-i) set to 95%. A maximum likelihood phylogenetic tree was constructed using RAxML-NG v1.1.0 [42] with the GTR+G model with 1,000 bootstrap replicates. FigTree v1.4.4 (http://tree.bio.ed.ac.uk/software/figtree/) was used to visualize and re-root the trees. Single nucleotide polymorphism (SNP) calculation involved 1,892 core genes that were used to build the *V. cholerae* tree. Multiple nucleotide polymorphisms (MNPs) greater than one nucleotide and indels were excluded from the final SNP counts. For *V. cholerae*, closely related reference strains for Clade1 and Clade2 were selected using sequence type information and PopPUNK lineage information from Vibriowatch database under PathogenWatch (https://pathogen.watch/). The detailed information of reference strains is available in S4 Table.

For phylogenetic analysis of the toxin coregulated pilin (*tcpA*) gene, the *tcpA* gene was first extracted from each strain using BLAST results, only including those with a BSR ≥ 0.4. Strain Cam5 was considered lacking *tcpA* and strain Cam14 having a truncated *tcpA* with a huge indel at the C-terminal, thus were excluded from the *tcpA* phylogeny analysis. To ensure a comprehensive analysis, reference strains with various coverage and identity values were incorporated. Extracted genes were aligned using MUSCLE v5.1.0 [43], and the tree was constructed following the same process as the core genome trees.

## Core genome multilocus sequence typing (cgMLST) analysis

cgMLST analysis was done on 260 *V. cholerae* isolates, including 19 Cameroon environmental strains and reference strains closely related to the Cameroon strains from Clade1 and Clade2.

From 323 *V. cholerae* environmental strains with available cgMLST data in the pubMLST database (https://pubmlst.org/organisms/vibrio-cholerae), *V. paracholerae* strains and PG clade environmental strains were excluded to prevent biases. The network analysis was done using a R package igraph [44], with133 allelic differences as a sublineage threshold [28]. Visualization was done with Cytoscape [45].

## Results/Discussion

### Isolation of *tcpA*+ non-O1/O139 *V. cholerae* (NOVCs) and *V. paracholerae* from freshwater sources in Cameroon

A total of 19 *V. cholerae* and one *V. paracholerae* were isolated from freshwater sources in North Cameroon. A core genome phylogenetic analysis was performed to explore potential linkages between our isolates and reference strains. The *viuB* genotypes of the Cameroon isolates were also identified, as they can distinguish subspecies lineages within *V. cholerae* and *V. paracholerae* populations [28]. Generally, each phylogenetic group exhibited a distinct *viuB* genotype (Fig 1), consistent with previous findings [25,28].

Several *V. cholerae* strains from Cameroon grouped into distinct clades with known reference strains, indicating potential evolutionary links (Fig 1). For instance, isolates Cam5, Cam7, Cam13, Cam14, Cam19, Cam21, and Cam22 were closely related and formed a monophyletic clade (Clade1 in Fig 1). Considering these strains were isolated from different geographical locations (Maroua and Bibemi) and at different times (April and May in 2018), they likely represent a common genotype in the region (S1 Fig and S1 Table). This clade displayed a close phylogenetic relationship with Kenyan environmental strains, all carrying *viuB*-25 and belonging to sequence type ST-538 (https://pathogen.watch/). Isolate Cam23 in Clade2 was closely related to NOVC Argentinian clinical strains, isolated during the 1990s cholera epidemic in Argentina [20,46]. It was also closely related to a non-O1/O139 clinical strain from Haiti, all sharing the *viuB*-33 genotype and sequence type ST-338. Comparing SNP counts among these strains across 1,892 core genes, the Cameroon strains from Clade1 and Clade2 showed much less genetic divergence with reference strains from other countries than what is typically seen between environmental strains from different geographical locations. For example, the Cam19 strain from Clade1 exhibited 218 SNPs relative to Kenyan environmental strain KNE18. Similarly, Cam23 from Clade2 showed 100 SNPs and 145 SNPs in comparison to the Argentinian clinical strain TUC_T2734 and Haitian clinical strain 2010V-1116, respectively. Other Cameroon strains were grouped into three distinct clades, none of which were closely related to strains from other countries (Fig 1).

In addition to *V. cholerae*, one *V. paracholerae* strain (Cam10) was isolated from Cameroon stream water. This is the first official report of *V. paracholerae* from Sub-Saharan Africa. *V. paracholerae* is the closest sister species of *V. cholerae*, often misidentified as the latter due to their high genetic similarity [25]. However, digital DNA-DNA hybridization (dDDH) and phenotypic differences confirmed that *V. paracholerae* is a separate species from *V. cholerae* [25]. *V. paracholerae* lacks cholera toxin (CTX) crucial for causing cholera, but possesses other virulence factors found in *V. cholerae*, such as repeat-in-toxin (RTX) cluster, haemolysin (*hlyA*), cholera toxin transcriptional activator (*toxR*), and outer membrane protein *ompU* [26]. Phylogenetic analysis indicates the global presence of *V. paracholerae* from both clinical and environmental sources (S2 Fig), including clinical sources in the United States, Yemen [47] and Mozambique as well as natural waters in cholera-endemic area like Algeria, Cameroon, and Bangladesh (S4 Table). *V. paracholerae* isolated in Cameroon was encoding *viuB*-5, which is a *viuB* type associated with the *V. paracholerae* species [25] (S2 Fig).

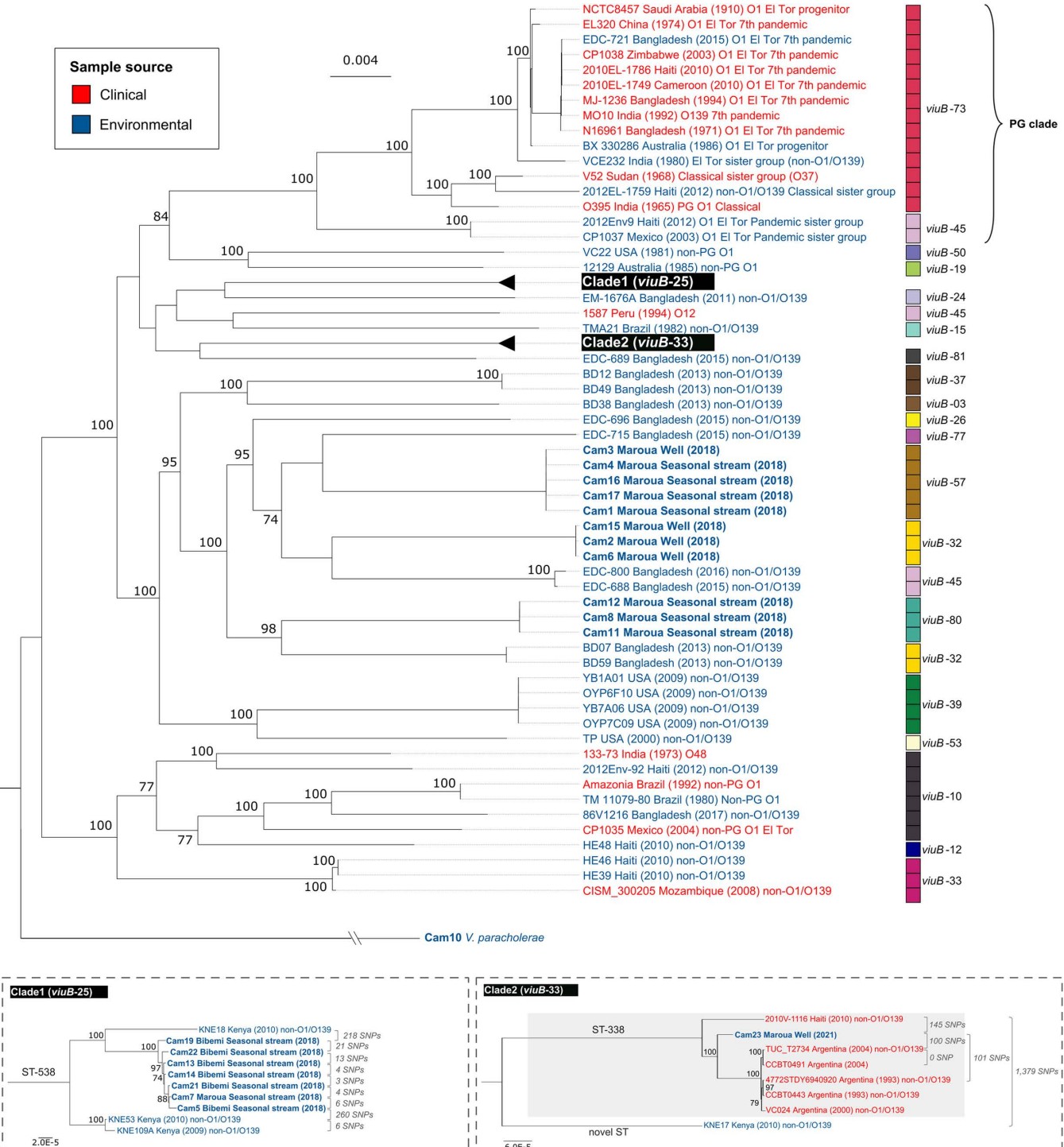

**Fig 1. Phylogenetic tree of *V. cholerae* strains.** Maximum-likelihood (ML) tree was constructed using RAxML-NG v. 1.1 with GTR+G model, using 1,892 core genes. The *viuB* genotype of each strain is indicated on the right side of the tree. Each strain is color-coded by sample source, with clinical strains in red and environmental strains in blue. Strains from this study are highlighted in bold. The tree is rooted with *V. paracholerae* strain Cam10 as an outgroup. Bootstrap values from 1,000 replicates are shown at the nodes, with only values greater than 70 indicated. The scale bar represents nucleotide substitutions per site. Subtrees for Clade1 (*viuB*-25) and Clade2 (*viuB*-33) were constructed with 3,260 and 3,253 core genes, respectively, with single nucleotide polymorphisms (SNPs) counts noted beside the tree leaf. ST: sequence type using seven housekeeping genes; PG clade: Pandemic-generating clade.

BLAST atlas analysis of NOVC Cameroon strains against *V. cholerae* reference strain N16961 revealed that several pathogenicity islands, including CTX-Phi cluster, Vibrio pathogenicity island-I (VPI-I), Vibrio pathogenicity island-II (VPI-II), and Vibrio seventh pandemic island-II (VSPI-II), were partially present (S3 Fig). To further evaluate the conservation of these virulence factors, we implemented blast score ratios (BSR) (37), revealing significant diversity in the virulence factors among Cameroon environmental strains (Fig 2). None of the Cameroon isolates harbored the cholera toxin (CTX) gene. However, the Cam23 strain contained other virulence genes encoded within the core region of the CTX cluster: *ace* (VC1459) gene encoding accessory cholera enterotoxin that stimulates ion transporters in intestinal epithelial cells that causes fluid secretion [48], and *zot* (VC1458) gene encoding zonula occludens toxin that is essential for CTX morphogenesis and increasing intestinal epithelial permeability

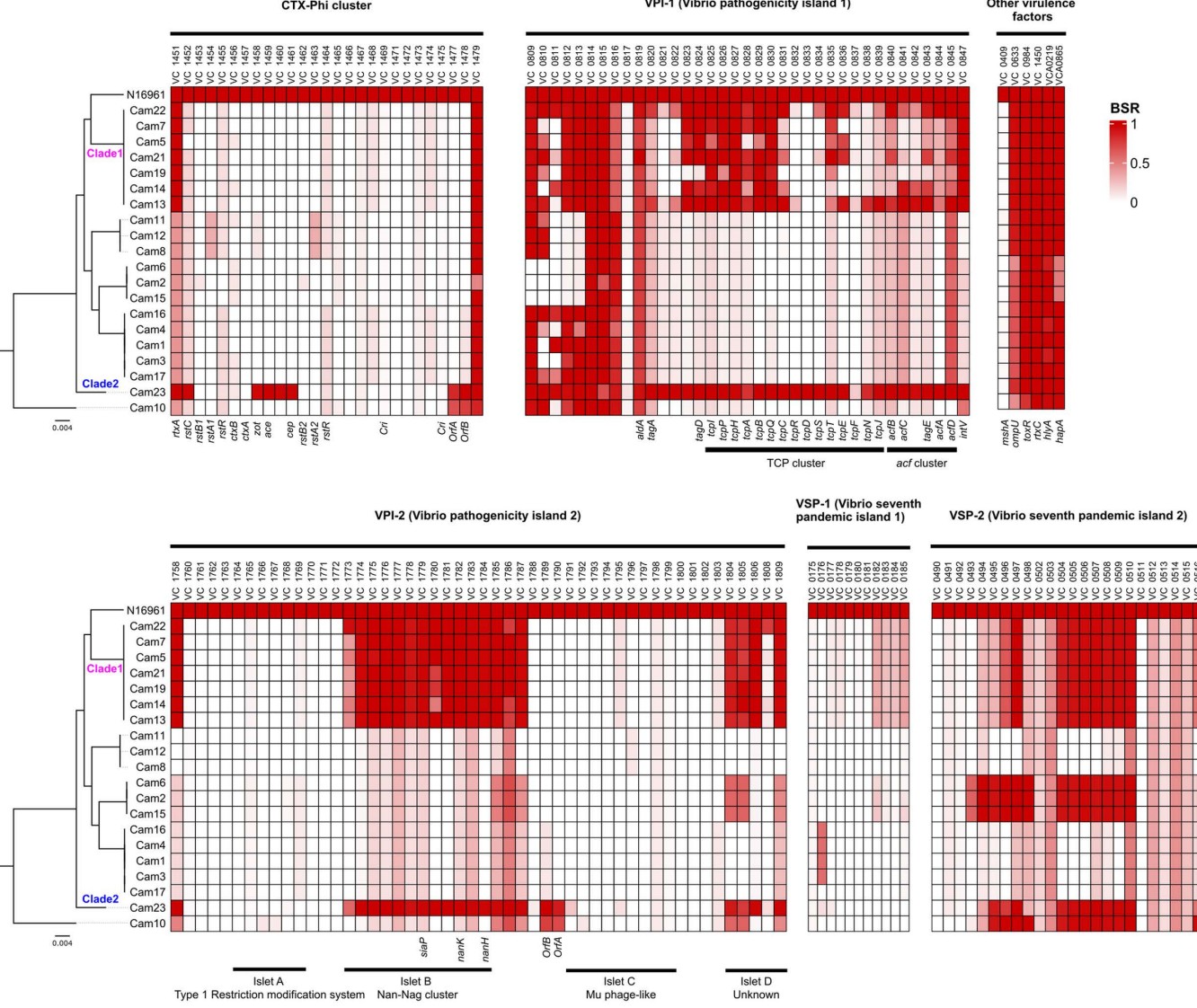

**Fig 2. Heatmap of the virulence gene profile of *V. cholerae* and *V. paracholerae* strains isolated from Cameroon waters.** For each gene, presence and similarity are indicated by a color gradient from white (absent) to dark red (identical), according to blast score ratio (BSR). Each row represents a different strain, with *V. cholerae* El Tor strain N16961 at the top as a reference.

[49,50]. *V. paracholerae* strain Cam10 lacked most virulence factors present in the reference strain N16961 but encoded a highly similar homolog of phage integrase (VC0516) (BSR ~ 0.95). Additionally, the RTX (repeat in toxin) gene cluster, known as a virulence factor in non-toxigenic *V. cholerae* [51], was partially present in the Cameroon strains. The gene *rtxC* (VC1450) was highly conserved across all strains (BSR > 0.99), whereas *rtxA* (VC1451) was found in nearly all strains, except *V. paracholerae* Cam10. This finding is unexpected, as previous studies have shown that *rtxC* is more commonly detected in NOVCs than *rtxA* [22,52].

Moreover, the RS regions of the CTX gene cluster were absent in all strains, except for Cam23, which contained a homolog of *rstC* (VC1452). RS regions encode genes responsible for phage replication (*rstA*), integration (*rstB*), regulation (*rstR*), and antirepressor (*rstC*), though the genetic components and arrangement differ across strains [53]. Transposase genes (VC1477 and VC1478) were present in Cam23 and *V. paracholerae* Cam10. Other toxin genes including *toxR*, *hylA*, *ompU*, and *hapA* were also detected in all strains, reflecting their virulence potential. Some encoded *ompU* and *hapA* variants with BSRs ranging from 0.61 to 0.77. These virulence factors are known to contribute to the virulence traits in nontoxigenic *V. cholerae* [51].

Genomic regions of the VPI-2 island, which are typically found in toxigenic O1 strains [54], are grouped into smaller DNA regions known as islets, each potentially encoding virulence factors. These islets include Islet A (Type 1 Restriction modification system, VC1764 to VC1769), Islet B (Nan-Nag cluster, VC1773 to VC1784), Islet C (Mu phage-like region, VC1791 to VC1799), and Islet D with unknown function (VC1804 to VC1809) [55]. Islet A and Islet C were absent in all Cameroon strains (Fig 2). Islet B was variably present, being either fully or partially present in strains from Clade1 and Clade2. Prior studies have reported that NOVCs causing gastroenterological diseases often encode a truncated VPI-2 with the Nan-Nag gene cluster [52,56]. This cluster is crucial for the pathogenicity of toxigenic *V. cholerae*, as it encodes proteins for sialic acid transport and catabolism that facilitate sialic acid scavenging, uptake, and catabolism [56,57], the latter being a common molecule in vertebrate cells. It also produces accessory virulence factor neuraminidase (*nanH*), which converts oligosaccharides to ganglioside GM1-like molecules on host cell surfaces, thereby enhancing the penetration of cholera toxin [54,57].

Although VSP gene elements are widely found in strains associated with the 7th cholera pandemic, their specific role in infection remains unclear. Similar to previous finding about environmental NOVC strains in German coastal waters (22), some of the Cameroon strains were also found to encode homologs of the VSP-2 region genes VC0494-VC0498 and VC0504-VC0510. The presence of VSP-2 elements in both pandemic and environmental strains suggests their role in environmental niche adaptation and enhancing pathogenicity [58,59].

Interestingly, noncanonical toxin co-regulated pilus (TCP) with a highly conserved *tcpA* (VC0828) gene were present in the Cameroon strains from Clade1 and Clade2. Unlike O1/O139 pandemic strains, most environmental NOVC strains generally do not encode cholera toxin and VPI genes, including *tcpA* [23,51,60,61]. The *tcpA* gene encodes a major structural protein of TCP, which enables the effective colonization of the human gut. It also serves as a receptor for filamentous phage CTXφ, which carries cholera toxin, thus converting non-toxigenic (*ctx*-) strains into toxigenic (*ctx*+). Notably, Cam23 also harbored a complete accessory colonizing factor (*acf*) cluster adjacent to the TCP cluster. The *acf* cluster is required for full virulence and works synergistically with the TCP gene cluster to enhance the colonization capabilities of the bacteria [62].

Genes associated with antibiotic resistance were detected among several NOVC strains in Cameroon, as detailed in S2 Table. 52.6% (10/19) of the strains encoded blaCARB-7 gene,

likely providing resistance to ampicillin. This resistance gene was observed in specific Cameroon clades encoding *viuB*-57, *viuB*-32, or *viuB*-80. Ampicillin resistance in *V. cholerae* has frequently been reported in strains from both environmental and diarrheal stool samples [63,64]. Other Cameroon strains exhibited resistance genes against antibiotics commonly used for cholera treatment. For instance, Cam1, Cam16, and Cam17 strains were found to carry *qnrVC* gene, conferring resistance to ciprofloxacin. Reduced susceptibility to ciprofloxacin has been reported in *V. cholerae* across Asia and Haiti, and since 2010, these strains have also emerged in Western Africa, including Nigeria and Cameroon [65,66]. Furthermore, strains from Cameroon Clade1 encoded *catB9* gene, which provides resistance against chloramphenicol. There has been a sharp increase in resistance to ampicillin and chloramphenicol since the early 1990s, which is likely associated with their widespread usage [67].

Our results show the genomic diversity of *V. cholerae* and *V. paracholerae* isolated from Cameroon waters, while also revealing shared virulence factors among the strains. Particularly, the *tcpA* gene was highly conserved among strains in Clade1 and Clade2 (Fig 1), suggesting the potential direct or indirect (*via* other bacteria) HGT with pandemic strains.

## Potential human carriage of NOVCs and transfer of NOVCs across countries and continents

Contrary to the well-known transmission capability of O1/O139 strains, nontoxigenic environmental NOVC strains outside the pandemic lineage are generally believed to lack the ability to disseminate across different geographical locations [68]. However, core genome phylogeny (Fig 1) shows the close relatedness of NOVC Cameroon strains from Clade1 and Clade2 with strains found in distant regions such as Kenya, Haiti, and Argentina. Subsequently, we compared their virulence profiles and sublineage classifications.

NOVC Cameroon strains from Clade1 showed noticeable variations in virulence profiles compared to their closely related Kenyan environmental reference strains (S4 Fig). While the latter showed nearly complete TCP and *acf* clusters, which were almost identical to the N16961 strain, Cameroon strains showed less conservation in these regions. Most Cameroon strains lacked many loci from both TCP cluster (VC0832-VC0834, VC0837-VC0839) and *acf* cluster (VC0840-VC0844). However, Cam22 and Cam13 showed a full-length *acf* cluster, with some divergence from the reference N16961 strain (S3 Table). Both Cameroon Clade1 strains and Kenyan environmental strains were missing most loci from the CTX-Phi cluster. Additionally, the *tcpA* gene was present in most strains, except for Cam14 and Cam5 (BSR < 0.4).

Cam23 from Clade2 showed virulence profile closely resembling those of clinical NOVC strains from Haiti and Argentina, especially Argentinian clinical strain 4772STDY6940920 (S5 Fig). The latter shared nearly identical virulence profiles with Cam23, except for the additional genes VC1466 and VC1472 (both encoding hypothetical protein) present only in 4772STDY6940920. A noncanonical TCP pattern was also observed in these strains. Although most TCP genes were highly conserved, *tcpF* (VC0837) was absent in all the strains in Clade2 (BSR < 0.4). Kenyan environmental strain KNE17, closely related to Clade2 (Fig 1), also showed a similar virulence profile to other strains from Clade2, but it encoded *tcpA* (VC0828) variant (BSR ~ 0.69) that is genetically distinct from *tcpA* in Clade1 and Clade2. It also encoded both *ctxA* and *ctxB*, suggesting its toxigenic potential [14].

Sublineage, usually corresponding to the 7 genes traditional MLST sequence type (ST), are widely used as the taxonomic unit to look at dispersal of a pathogenic species across the world. When this is done, bacteria are considered to have dispersed geographically if the same ST is found in different countries [68,69]. To determine the sublineages of the Cameroon *V. cholerae* strains, a network analysis was conducted based on core genome multilocus sequence typing

(cgMLST), using 133 allelic differences as a sublineage threshold, which has been shown to be roughly equivalent to traditional MLST STs [68]. We included all NOVC strains with available cgMLST data. The result clearly showed a connection between Cameroon environmental strains and reference strains from Kenya and Argentina (Fig 3). With the exception of two Cameroon clusters and strain 692-79 from the US and 857 from Bangladesh, other strains in the same cgMLST sublineage shared a country of origin, aligning with previous findings [68]. Consistent with our phylogenetic analysis, Cam23 from Clade2 grouped within the same cgMLST sublineage as non-O1 clinical strains from Argentina. Likewise, strains from Clade1 grouped with Kenyan environmental strains. This suggests that Cameroon NOVC strains from Clade1 and Clade2 may have spread over long distances by human carriers. Although a *ctx-/tcpA+* Haitian clinical strain 2010V-1116 and a *ctx+/tcpA+* Kenyan environmental strain KNE17 showed close phylogenetic relationships to the Cam23 strain (Fig 1), they were not grouped into the same sublineage with any other strains from Clade2 (Fig 3).

Particularly, Argentinian clinical strains closely related to Cam23 belong to the A3 clade of non- 7th pandemic El Tor (7PET) strains, which coexisted with the 7PET LAT-1 lineage responsible for cholera epidemics in Argentina [20]. The A3 clade has been thought to be composed solely of clinical strains endemic to Argentina. However, our phylogenetic and cgMLST analyses indicate that Cam23 environmental strain also belongs to this A3 clade. Notably, both Argentinian clinical strains and Cam23 possess the same unique genetic elements from the uncommon T3SS-2β system [70], which is believed to have originated from clinical sources [20]. This suggests an evolutionary and epidemiological link between the Argentinian clinical strains and Cam23. Furthermore, the detection of shared virulence elements raises the possibility that larger scale outbreaks associated with this clade may have gone undetected.

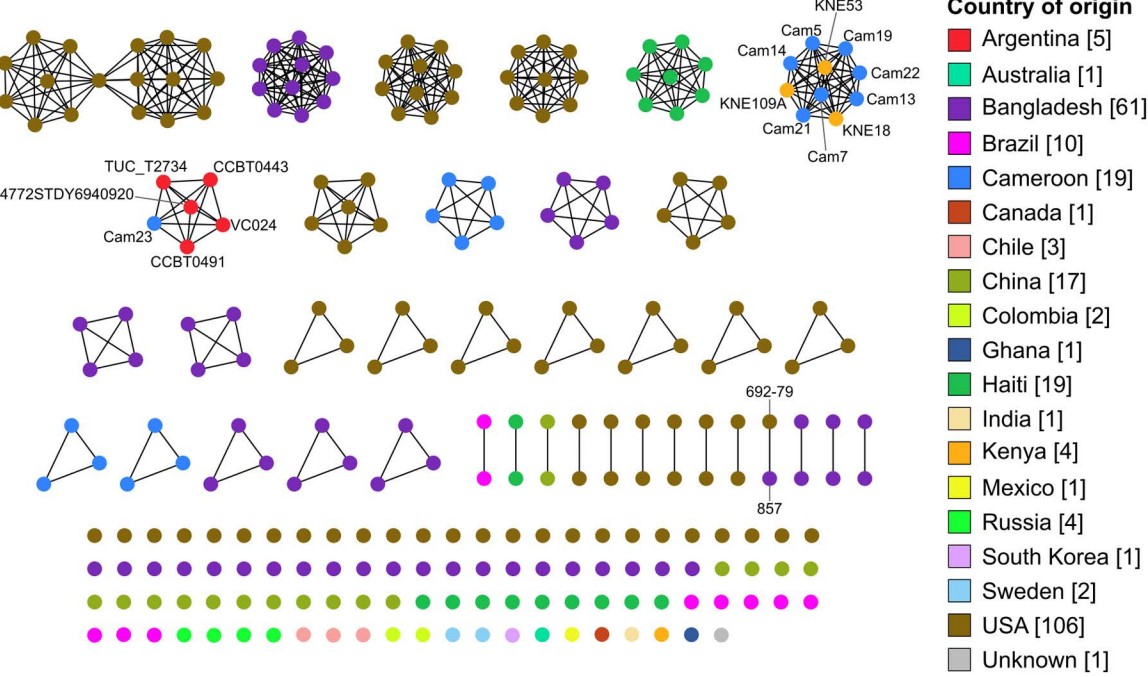

**Fig 3. Network analysis of 260 *V. cholerae* strains based on core genome MLST (cgMLST) profiles.** Nodes represent individual strains, color-coded based on their country of origin. Strains from the same sublineage (allelic difference equal to or less than 133) are linked together. Strains from Clade1 and Clade2, and strains from different countries that cluster together are labeled.

Considering the SNP acquisition rate of 3.3 SNPs per genome per year in the O1 7th pandemic [71], we can extrapolate from 218 SNPs difference between Cam19 strain from Clade1 and Kenyan environmental strain KNE18, that these strains diverged from their common ancestor approximately 66 years ago. Similarly, with a difference of 101 SNPs, Cam23 from Clade2 and Argentinian clinical strain 4772STDY6940920 likely diverged ~31 years ago, roughly matching the difference in their isolation years (Fig 1). This is comparable to divergence between strains inside the El Tor or Classical pandemic lineages, which have been clearly shown to originate in Asia and dispersed to other continents by human carriers [72]. However, it is important to note that molecular evolution rates can vary across different lineages [73].

Together, phylogenetic analysis, virulence profile, and cgMLST results indicate a possible transmission of environmental NOVCs across different geographical regions, across both countries and continents. While it is well known that O1/O139 strains in the pandemic lineage are capable of human-mediated long-distance transmission, our findings suggest that such transmissions are not limited to O1/O139 strains. To our knowledge, this study presents the first evidence of potential transmission of nontoxigenic NOVC environmental strains across different continents. Previous research explained the global spread of *ctx*-/ *tcpA*+ O1 *V. cholerae* strains that have caused multiple cholera outbreaks worldwide [74]. It is thought that the *tcpA* gene, which facilitates the colonization of the human gut, has given these strains pathogenic potential. Considering that most Cameroon strains from Clade1 and Clade2 encode *tcpA*, it is plausible that these lineages were disseminated with the help of human hosts, then later went through genetic recombination and mutations.

## Horizontal gene transfer of *tcpA* between Cameroon NOVCs and the Classical O1 clade

BLAST analysis performed on the Cameroon genomes revealed that Cam23 from Clade2 and most strains from Clade1 encode a *tcpA* gene. A phylogenetic analysis reveals that *tcpA* genes from Cameroon strains in Clade1 formed a distinct group, named "*tcpA* Cameroon Clade1" (Fig 4). This clade closely resembles the *tcpA* Classical type gene from the V52 Classical sister strain (18), differing by only four SNPs and a single insertion at the last position (G -> GC). Considering a mean ANI of 98.15% between Clade 1 strains and the V52 Classical sister strain, an expected divergence would be 13 SNPs across the 675 bp *tcpA* gene. However, only four SNPs were observed, much fewer than expected. Of these four SNPs, three were synonymous mutations, not affecting the amino acid sequence, while one SNP at position 605 (C -> A) resulted in a substitution from threonine (Thr) to lysine (Lys) at amino acid position 202. The insertion at the terminal position introduced an additional serine (Ser) residue, which is also present in N16961. Some variations were observed within the *tcpA* O1 Classical type, including an amino acid substitution at position 209 from lysine (Lys) to glutamate (Glu).

This close similarity between *tcpA* Cameroon Clade1 and *tcpA* from the O1 Classical lineage implies horizontal gene transfer of the *tcpA* gene between NOVC strains and pandemic Classical O1 strains [75]. A previous study has also reported the existence of *tcpA* gene similar to the *tcpA* classical type in environmental NOVC strains from Calcutta, India [19]. Intriguingly, while the Haitian clinical strain 2010V-1116 was genomically related to Cam23 and the Argentinian clinical strains from Cameroon Clade2 as shown in Fig 1, it encoded a *tcpA* gene from the *tcpA* Cameroon Clade1 (Fig 4), with an insertion at the last position (G -> GC). This implies the transfer of the *tcpA* gene from Clade1 to the Haitian strain, suggesting possible co-infections with different strains, as was previously observed in Haiti [21]. Furthermore, other environmental strains from Haiti and Korea also encoded the O1 *tcpA* classical type (**Fig 4**), highlighting frequent HGT and complex evolution for this gene.

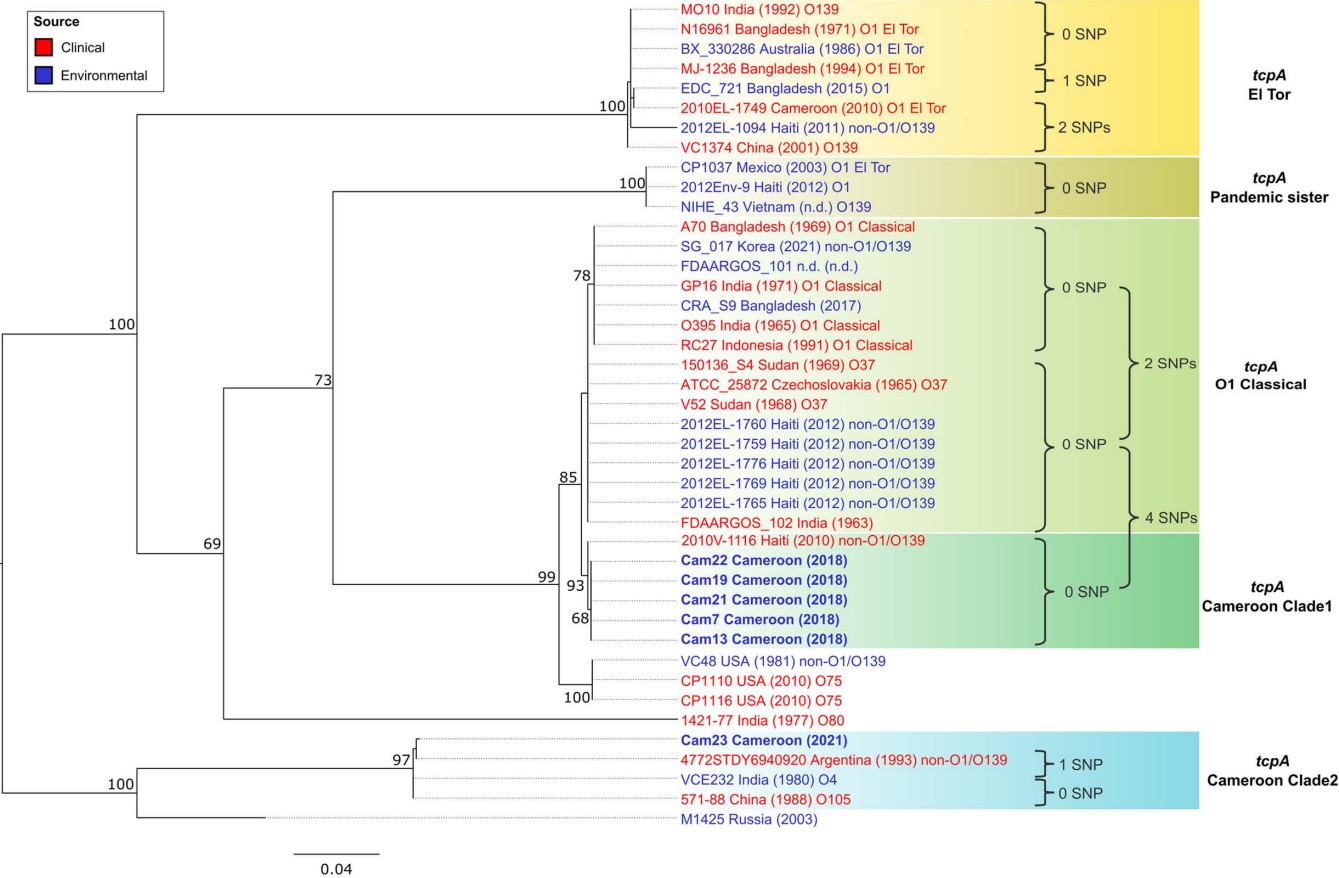

**Fig 4. Phylogeny of *tcpA* (toxin-coregulated pilus subunit A) gene in *V. cholerae*.** Maximum-likelihood (ML) tree was generated by RAxML-NG v. 1.1 based on the multiple sequence alignment of *tcpA* genes. Clinical and environmental strains are labeled in red and blue, respectively. Strains from this study are highlighted in bold. The tree is rooted with strains distant from the pandemic clade. Nodes are labeled with bootstrap support from 1,000 replicates. The scale bar represents nucleotide substitutions per site, and the SNP differences between the lineages are indicated beside the tree.

In the case of the *tcpA* gene of Cam23 strain from Clade2, it clustered in a distinct clade named "*tcpA* Cameroon Clade2", including clinical strains from Argentina and China, as well as an O4 environmental strain from India (**Fig 4**). The *tcpA* gene in Cam23 was nearly identical to that in the Argentinian clinical strain 4772STDY6940920 from non-7PET A3 clade, except one insertion at the last position (G->GC), leading to an additional serine (Ser) residue, a pattern observed in the Cameroon strains from Clade1 and 2010V-1116 from Haiti. This supports the hypothesis of a possible transfer of the A3 clade from Africa to Latin America, as suggested in a previous study [71], enabling broader geographical spread and exchange of virulence factors.

Overall, the presence of the *tcpA* gene in Cameroon NOVC strains and its close relationship with the *tcpA* O1 Classical type suggests an evolutionary link between these environmental strains and pandemic strains. The detection of similar *tcpA* sequences in strains in diverse geographical locations underscores the role of this gene in environmental adaptation and enhanced virulence capabilities of *V. cholerae*. These findings highlight the potential of NOVC strains to influence the genetic evolution of *V. cholerae* through its association with pandemic strains and vice versa.

A limitation of this study is the restricted number of samples available from Africa, which limits the ability to fully assess the prevalence of these NOVC lineages in the region. Furthermore, the study relied solely on environmental samples; clinical strains from local cases would be required to establish more comprehensive epidemiological linkages among these lineages

and their potential role in local cholera transmissions. Future studies should aim to collect a larger number of strains, including both environmental and clinical samples, to better understand the prevalence and transmissions of these lineages in the region.

## Conclusions

The ability of toxigenic O1 and O139 *V. cholerae* to infect humans and spread over long distances is not exclusive to these biotypes but likely also found in NOVC strains. Our findings show that NOVC strains from Cameroon waters, with highly conserved *tcpA* genes, may have spread across different geographical regions with the help of human hosts. The co-circulation of NOVC with epidemic strains during cholera outbreaks and their shared virulence factors underscores the need to reevaluate the role of environmental strains in the evolution and transmission of *V. cholerae*. This research emphasizes the importance of expanding genomic surveillance in Africa, particularly to include environmental strains, to enhance our understanding of the dynamics of cholera outbreaks. Further investigations are necessary to determine the prevalence of these NOVC lineages in Northern Cameroon region and their roles in local cholera outbreaks.

## Supporting information

**S1 Fig. Map of North Cameroon showing sampling sites.** The base map was adapted from Diva-Gis (https://diva-gis.org/data.html). The license information is available at https://en.wikipedia.org/wiki/DIVA-GIS.
(TIF)

**S2 Fig. Phylogenetic tree of V. paracholerae strains.** Maximum-likelihood (ML) tree was generated by RAxML with GTR+G model, using 2,323 core genes. viuB genotype of each strain is indicated on the right side of the tree. Each strain is color-coded by sample source, with clinical strains in red, environmental strains in blue, and one strain (07-2425) from unknown source in black. Strain Cam10 from this study is highlighted in bold. The tree is rooted with *V. cholerae* strain N16961 as an outgroup. Nodes are labeled with bootstrap support from 1,000 replicates. The scale bar represents the number of nucleotide substitutions per site. n.d.: no date (collection date unknown).
(TIF)

**S3 Fig. BLAST atlas comparison of 20 V. cholerae and one V. paracholerae strains against N16961 reference strain.** Genetic similarities across two chromosomes of V. cholerae and V. paracholerae strains are shown. The reference N16961 strain is depicted as the blue backbone, with forward and reverse strands separately. Gene regions with BLAST hits are colored according to the identity scores, while the blank regions indicate no hits. Primary pathogenicity islands and superintegron in N16961 are annotated around the circles. Strains are arranged from *V. cholerae* Cam23 on the outer circle to *V. paracholerae* Cam10 in the innermost circle, following the legend at the right of the plot.
(TIF)

**S4 Fig. Heatmap of virulence gene profile of V. cholerae strains from Clade1.** For each gene, presence and similarity are indicated by a color gradient from white (absent) to dark red (identical), according to the blast score ratio (BSR). Each row represents a different strain, with *V. cholerae* El Tor strain N16961 at the top as a reference. The strains from this study are highlighted in blue.
(TIF)

**S5 Fig. Heatmap of virulence gene profile of V. cholerae strains from Clade2.** For each gene, presence and similarity are indicated by a color gradient from white (absent) to dark

red (identical), according to the blast score ratio (BSR). Each row represents a different strain, with *V. cholerae* El Tor strain N16961 at the top as a reference. The strains from this study are highlighted in blue.
(TIF)

**S1 Table. Genomic characteristics of assembled genomes.** ANI was compared with the closest species, either *V. cholerae* (GCF_008369605.1) or *V. paracholerae* (GCF_003311965.1) reference genome. Other genomic statistics are included: genome length, the number of contigs, N50, length of the longest and shortest contigs, GC content, average sequencing coverage, total number of coding sequences (CDS), rRNA, and tRNA genes.
(XLSX)

**S2 Table. Antimicrobial resistance (AMR) prediction and sequence types (ST) of assembled genomes.**
(XLSX)

**S3 Table. Blast score ratio (BSR) results.**
(XLSX)

**S4 Table. Reference strains used in this study.**
(XLSX)

## Acknowledgments

We thank Moussa Djaouda's group for sample collection and sample processing. We thank SCELSE sequencing Team for library preparation and sequencing.

## Author contributions

**Conceptualization:** Deborah Yebon Kang, Mohammad Tarequl Islam, Roméo Wakayansam Bouba, Zoua Wadoubé, Moussa Djaouda, Yann Felix Boucher.

**Data curation:** Deborah Yebon Kang, Mohammad Tarequl Islam, Zoua Wadoubé.

**Formal analysis:** Deborah Yebon Kang, Mohammad Tarequl Islam.

**Investigation:** Deborah Yebon Kang, Mohammad Tarequl Islam, Roméo Wakayansam Bouba, Zoua Wadoubé, Moussa Djaouda, Yann Felix Boucher.

**Methodology:** Deborah Yebon Kang, Mohammad Tarequl Islam, Roméo Wakayansam Bouba, Zoua Wadoubé, Moussa Djaouda, Yann Felix Boucher.

**Resources:** Deborah Yebon Kang, Mohammad Tarequl Islam, Roméo Wakayansam Bouba, Zoua Wadoubé, Moussa Djaouda.

**Supervision:** Yann Felix Boucher.

**Visualization:** Deborah Yebon Kang, Zoua Wadoubé.

**Writing – original draft:** Deborah Yebon Kang.

**Writing – review & editing:** Deborah Yebon Kang, Mohammad Tarequl Islam, Moussa Djaouda, Yann Felix Boucher.

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
