## [Decision Letter · Decision Letter 0]

28 Nov 2024

PNTD-D-24-01435Non-O1/O139 environmental Vibrio cholerae from Northern Cameroon reveals potential intra-/inter-continental transmissionsPLOS Neglected Tropical DiseasesDear Dr. Kang, Thank you for submitting your manuscript to PLOS Neglected Tropical Diseases. After careful consideration, we feel that it has merit but does not fully meet PLOS Neglected Tropical Diseases's publication criteria as it currently stands. Therefore, we invite you to submit a revised version of the manuscript that addresses the points raised during the review process. Please submit your revised manuscript within 30 days Jan 27 2025 11:59PM. If you will need more time than this to complete your revisions, please reply to this message or contact the journal office at plosntds@plos.org. Please include the following items when submitting your revised manuscript: * A rebuttal letter that responds to each point raised by the editor and reviewer(s). You should upload this letter as a separate file labeled 'Response to Reviewers '. This file does not need to include responses to any formatting updates and technical items listed in the 'Journal Requirements' section below. * A marked-up copy of your manuscript that highlights changes made to the original version. You should upload this as a separate file labeled 'Revised Manuscript with Track Changes '. * An unmarked version of your revised paper without tracked changes. You should upload this as a separate file labeled 'Manuscript '. If you would like to make changes to your financial disclosure, competing interests statement, or data availability statement, please make these updates within the submission form at the time of resubmission. Guidelines for resubmitting your figure files are available below the reviewer comments at the end of this letter. We look forward to receiving your revised manuscript. Kind regards, Benedikt Ley, PhDGuest EditorPLOS Neglected Tropical DiseasesMathieu PicardeauSection EditorPLOS Neglected Tropical Diseases 

Shaden Kamhawi

co-Editor-in-Chief

Paul Brindley

co-Editor-in-Chief

**Journal Requirements:**

2) Some material included in your submission may be copyrighted. According to PLOSu2019s copyright policy, authors who use figures or other material (e.g., graphics, clipart, maps) from another author or copyright holder must demonstrate or obtain permission to publish this material under the Creative Commons Attribution 4.0 International (CC BY 4.0) License used by PLOS journals. Please closely review the details of PLOSu2019s copyright requirements here: PLOS Licenses and Copyright. If you need to request permissions from a copyright holder, you may use PLOS's Copyright Content Permission form.

Potential Copyright Issues:

- Figure S1. Please provide a direct link to the base layer of the map (i.e., the country or region border shape) and ensure this is also included in the figure legend; and provide a link to the terms of use / license information for the base layer image or shapefile. We cannot publish proprietary or copyrighted maps (e.g. Google Maps, Mapquest) and the terms of use for your map base layer must be compatible with our CC BY 4.0 license.

3) Please amend your detailed Financial Disclosure statement. This is published with the article. It must therefore be completed in full sentences and contain the exact wording you wish to be published.

**Reviewers' comments:**

Reviewer's Responses to Questions

**Key Review Criteria Required for Acceptance?**

**Methods**

-Are the objectives of the study clearly articulated with a clear testable hypothesis stated?

-Is the study design appropriate to address the stated objectives?

-Is the population clearly described and appropriate for the hypothesis being tested?

-Is the sample size sufficient to ensure adequate power to address the hypothesis being tested?

-Were correct statistical analysis used to support conclusions?

-Are there concerns about ethical or regulatory requirements being met?

Reviewer #1: All of the above points (1-5) can be answered with yes.

There are no concerns regarding compliance with ethical or legal requirements.

Reviewer #2: -Are the objectives of the study clearly articulated with a clear testable hypothesis stated?

In the last paragraph of the introduction, the authors give an overview of the results and the conclusion. Instead, they should clearly state the objective and hypothesis of their study here.

-Is the study design appropriate to address the stated objectives?

The objectives need to be more clearly stated, however the study design is appropriate.

-Is the population clearly described and appropriate for the hypothesis being tested?

The population/sampling is clearly described; however, the hypothesis/objective needs to be clarified.

-Is the sample size sufficient to ensure adequate power to address the hypothesis being tested?

This is a descriptive analysis and the sample size is sufficient for this kind of analysis.

-Were correct statistical analysis used to support conclusions?

Yes, the descriptive analysis supports the manuscript’s conclusions.

-Are there concerns about ethical or regulatory requirements being met?

The authors need to add an ethics statement in the methods section.

**Results**

-Does the analysis presented match the analysis plan?

-Are the results clearly and completely presented?

-Are the figures (Tables, Images) of sufficient quality for clarity?

Reviewer #1: All of the above points can be answered with yes.

Reviewer #2: -Does the analysis presented match the analysis plan?

The authors should state whether or not the analyses were based on a prospective analysis plan.

-Are the results clearly and completely presented?

Indeed, the results are very clearly and comprehensively presented within text, figures and supplementary data.

-Are the figures (Tables, Images) of sufficient quality for clarity?

The figures are of excellent quality.

**Conclusions**

-Are the conclusions supported by the data presented?

-Are the limitations of analysis clearly described?

-Do the authors discuss how these data can be helpful to advance our understanding of the topic under study?

-Is public health relevance addressed?

Reviewer #1: All of the above points can be answered with yes.

Reviewer #2: -Are the conclusions supported by the data presented?

Most conclusions are supported by the data, and very well laid out, however as these are only environmental samples the authors should consider to slightly tone down the conclusion that the presence of tcpA alone indicates infection/colonization potential and underline that they would need to collect clinical samples from patients to show this.

-Are the limitations of analysis clearly described?

No, these need to be added to the manuscript.

-Do the authors discuss how these data can be helpful to advance our understanding of the topic under study?

The authors explain very well how their findings may influence the understanding of expansion of NOVC strains and interaction with O1-strains.

-Is public health relevance addressed?

Public health relevance is addressed in the conclusion.

**Editorial and Data Presentation Modifications?**

Reviewer #1: Minor Revision

Overall, the manuscript is very well written and structured. Therefore, I only have minor corrections to suggest:

Page 5: “Water sample collection and isolation of V. cholerae“ – Please specify in this section how much water was filtered for a sample.

Page 6: “DNA extraction & sequencing“ last sentence in the first paragraph - Please do not start a sentence with an abbreviated gene name. Please check this for the entire manuscript.

Page 6: “DNA extraction & sequencing“ second paragraph - Please explain why were the two batches analyzed at different laboratories with different sequencing kits?

Page 7: “Genome assembly and species identification“ - Please explain using identities in % what corresponds to a perfect match of viuB to the reference gene and when it is a mismatch.

Page 12: “ of tcpA+ non-O1, non-O139 V. cholerae (NOVCs) and V. paracholerae from freshwater sources in Cameroon“ - Please always use lowercase letters for names, as ompU begins with a capital letter in the text on the start of page 12.

Reviewer #2: - Please use the names of cholera strains consistently, e.g. in the abstract non-O1/O139 is used, while in the introduction it’s non-O1/non O139.

- Introduction: “net increase in the impact of epidemics” -> Please clarify what you mean by impact.

- Results/Discussion:

o “(present in public databases but unpublished)” -> Please cite these databases.

o “Antibiotic resistance was observed among several NOVC strains in Cameroon“ -> Genes associated with antibiotic resistance were detected…

**Summary and General Comments**

Reviewer #1: Dear Authors,

Thank you for this detailed genetic comparison between O1/O139 V. cholerae and NOVC.

The authors discuss possible links between the association between O1/O139 V. cholerae and NOVC and their role in cholera transmission. In addition, the genetic virulence marker tcpA in NOVC, which shows similarities to that of classical O1 strains, suggests that these strains may play an important role in outbreak events and transmission. In addition, the authors found that non-O1/O139 V. cholerae strains from water sources in northern Cameroon are related to strains from Kenya and Argentina despite their genetic variability, suggesting cross-regional transmission. All these findings emphasize the need to consider non-O1/O139 strains in cholera surveillance, which may play a greater role in the spread of cholera than previously thought.

Reviewer #2: In this manuscript, Kang et al. describe a genetic analysis of non-O1/O139 V. cholerae environmental isolates from Northern Cameroon. They show that isolates from Northern Cameroon are closely related to strains from Kenya and Argentina, suggesting transmission inter-continental transmission. This has previously been shown only for toxigenic strains. Moreover, they show a close relation between original O1 and Cameroonian NOVC virulence factors, suggesting horizontal gene transfer between these strains. These results suggest that strengthening VC genomic surveillance, including environmental strains might help us better understand the dynamics of cholera epidemics. Thus, this study is important in its field of research, which aligns nicely with the major scope of PLOS NTDs.

The manuscript is excellently written, contains very well executed statistical analyses and is well reasoned throughout the discussion of the results. Most conclusions are based on the data generated by the authors and there are only a few points that need to be addressed before publication. One major limitation of the study is that only environmental samples have been included—it would have been very interesting to integrate clinical samples in this analysis! However, as these were not available, the authors need to adjust the interpretation of colonization/infection potential from their analysis of environmental samples (see above), and discuss this as a limitation of the study. Other minor concerns are listed above.

There is one more issue of this manuscript: A major goal of PLOS NTDs is to promote and profile the efforts of researchers in endemic countries in order to help build science in these regions (https://journals.plos.org/plosntds/s/journal-information). In HIC/LMIC research partnerships (such as the partnership behind this manuscript), power imbalances are well documented, and authorship criteria should be used inclusively to minimize parachute research and promote local researchers (https://doi.org/10.1136/bmjgh-2021-007632). Although this manuscript describes the epidemiology of Vibrio cholerae in Cameroon and samples were collected by a Cameroonian research team, only one researcher with a Cameroonian affiliation is named among the co-authors and in the acknowledgements. Please reconsider if there are any Cameroonian (junior) researchers who meet the PLOS NTDs authorship criteria (https://journals.plos.org/plosntds/s/authorship). That is, if there are researchers other than MD who contributed substantially to data acquisition (e.g. through sample collection) and agree to be personally accountable for their contribution, they should be given the opportunity to review and approve this manuscript to be included as co-authors. In this case, please get in touch with the PLOS NTDs editorial team to add them as co-authors. If after reconsideration you should insist that MD is the only Cameroonian researcher who contributed substantially to the submitted work, please explain why and list the Cameroonian researchers, who contributed but do not meet the authorship criteria by name in the acknowledgements (given they agree to be named).

PLOS authors have the option to publish the peer review history of their article (what does this mean? ). If published, this will include your full peer review and any attached files.

**Do you want your identity to be public for this peer review?** For information about this choice, including consent withdrawal, please see our Privacy Policy .

Reviewer #1: No

Reviewer #2: No

**Figure resubmission:** While revising your submission, please upload your figure files to the Preflight Analysis and Conversion Engine (PACE) digital diagnostic tool, https://pacev2.apexcovantage.com/. PACE helps ensure that figures meet PLOS requirements. To use PACE, you must first register as a user. Registration is free. Then, login and navigate to the UPLOAD tab, where you will find detailed instructions on how to use the tool. If you encounter any issues or have any questions when using PACE, please email PLOS at figures@plos.org. Please note that Supporting Information files do not need this step. If there are other versions of figure files still present in your submission file inventory at resubmission, please replace them with the PACE-processed versions.**Reproducibility:** To enhance the reproducibility of your results, we recommend that authors of applicable studies deposit laboratory protocols in protocols.io, where a protocol can be assigned its own identifier (DOI) such that it can be cited independently in the future. Additionally, PLOS ONE offers an option to publish peer-reviewed clinical study protocols. Read more information on sharing protocols at https://plos.org/protocols?utm_medium=editorial-email&utm_source=authorletters&utm_campaign=protocols

---

## [Editor Report · Decision Letter 1]

5 Feb 2025

Dear Ms Kang,

We are pleased to inform you that your manuscript 'Non-O1/O139 environmental Vibrio cholerae from Northern Cameroon reveals potential intra-/inter-continental transmissions' has been provisionally accepted for publication in PLOS Neglected Tropical Diseases.

Best regards,

Benedikt Ley, PhD

Guest Editor

Mathieu Picardeau

Section Editor

Shaden Kamhawi

co-Editor-in-Chief

Paul Brindley

co-Editor-in-Chief

---

## [Editor Report · Acceptance letter]

Dear Ms Kang,

We are delighted to inform you that your manuscript, "Non-O1/O139 environmental Vibrio cholerae from Northern Cameroon reveals potential intra-/inter-continental transmissions," has been formally accepted for publication in PLOS Neglected Tropical Diseases.

Best regards,

Shaden Kamhawi

co-Editor-in-Chief

Paul Brindley

co-Editor-in-Chief
